# Transcriptome Profiles of Leaves and Roots of Goldenrain Tree (*Koelreuteria paniculata* Laxm.) in Response to Cadmium Stress

**DOI:** 10.3390/ijerph182212046

**Published:** 2021-11-16

**Authors:** Qihao He, Tao Zhou, Jikang Sun, Ping Wang, Chunping Yang, Lei Bai, Zhiming Liu

**Affiliations:** 1College of Environmental Science and Engineering, Central South University of Forestry and Technology, Changsha 410004, China; hjheqihao@163.com; 2College of Life Science and Technology, Central South University of Forestry and Technology, Changsha 410004, China; 18684976822@163.com (T.Z.); sjks110@163.com (J.S.); leibaihbmu@163.com (L.B.); 3Guangdong Provincial Key Laboratory of Petrochemcial Process and Control, School of Environmental Science and Engineering, Guangdong University of Petrochemical Technology, Maoming 525000, China; 4College of Environmental Science and Engineering, Hunan University and Key Laboratory of Environmental Biology and Pollution Control (Hunan University), Ministry of Education, Changsha 410082, China; 5Department of Biology, Eastern New Mexico University, Portales, NM 88130, USA; Zhiming.liu@enmu.edu

**Keywords:** goldenrain tree, Cd stress, transcription factors, weighted gene co-expression network analysis, transcriptome

## Abstract

Cadmium (Cd) pollution is a widespread environmental problem. In this study, we explored the transcriptome and biochemical responses of goldenrain tree (*Koelreuteria paniculata* Laxm.) leaves and roots to Cd stress. Leaf and root growth decreased substantially under Cd stress (50 mg/L CdCl_2_), but leaf and root antioxidant mechanisms were significantly activated. In RNA-seq analysis, roots treated with 25 mg/L CdCl_2_ featured enriched GO terms in cellular components related to intracellular ribonucleoprotein complex, ribonucleoprotein complex, and macromolecular complex. In leaves under Cd stress, most differentially expressed genes were enriched in the cellular component terms intrinsic component of membrane and membrane part. Weighted gene co-expression network analysis and analysis of module–trait relations revealed candidate genes associated with superoxide dismutase (SOD), peroxidase (POD), and catalase (CAT) activities and malondialdehyde (MDA). Ten transcription factors responded to Cd stress expression, including those in C2H2, MYB, WRKY, and bZIP families. Transcriptomic analysis of goldenrain tree revealed that Cd stress rapidly induced the intracellular ribonucleoprotein complex in the roots and the intrinsic component of membrane in the leaves. The results also indicate directions for further analyses of molecular mechanisms of Cd tolerance and accumulation in goldenrain tree.

## 1. Introduction

Heavy-metal contamination in soil is widespread and poses serious environmental risks [1]. Heavy metals can be classified into two types according to their metabolic roles in plant development. One type is essential for plant growth but adversely affects development at excessive concentrations and includes copper, zinc, and iron. The other type is not essential and is hazardous to plant growth and includes cadmium (Cd) as one example [2]. Cadmium pollution is a widespread environmental problem [3]. It is released into soils primarily through wastewater irrigation and contaminated phosphate fertilizer [4]. A previous report showed that as much as 88 metric tons of cadmium was released into the environment in 1989 by the nonferrous metallurgy industry [5]. In industrial regions, the mean concentrations of Cd are 79.2 times greater than the Grade II environmental quality standard for soils in China (GB15618-1995) (about 20 mg/kg) [6].

Cadmium accumulation in plants may cause physiological, biochemical, and structural changes. Accumulation of cadmium can change the nutrient absorption of minerals, inhibit stomatal opening, disrupt Calvin cycle enzymes, photosynthesis, and carbohydrate metabolism, change antioxidant metabolism, and reduce crop productivity through interaction with plant water balance [7]. A large number of studies have shown that the main action site of Cd is the biosynthesis of photosynthetic pigments, especially chlorophyll and carotenoids, and plant growth [8,9].

Heavy-metal-contaminated soils must be decontaminated to reduce associated risks, maintain environmental health, and allow ecological restoration. Sodango et al. [10] proposed strategies to minimize Cd availability in soil and, thus, its uptake by crops. Physicochemical and biological techniques have been adopted to eliminate heavy metals from soils or reduce plant uptake of Cd, which include excavation, leaching, electroreclamation, application of microorganisms, and selection of low-Cd-accumulating plant cultivars [11,12,13]. Phytoremediation is an emerging technology in which plant uptake removes contaminants from the environment. It is a cost-effective and noninvasive alternative to conventional remediation approaches [14]. Types of phytoremediation include phytostabilization, phytostimulation, phytotransformation, phytofiltration, and phytoextraction [15].

The goldenrain tree *Koelreuteria paniculata* Laxm. (Sapindaceae) is widely distributed in Northern China. The goldenrain tree has traditionally been used for landscaping because of its beautiful yellow flowers. In recent years, some people have tried to use its seeds to produce biodiesel [16], which has high economic value. The tree has a deep root system, is salt- and drought-tolerant, and can survive short-term flooding. Recent studies on heavy-metal tolerance in goldenrain tree primarily investigated heavy-metal enrichment and corresponding physiological and biochemical changes, whereas few studies investigated genome-level changes. In this study, we explored changes in leaf and root transcriptomes, as well as biochemical activity, of goldenrain tree under Cd stress to explore its molecular response mechanism under Cd stress, as well as reveal its metabolic pathway and differential genes in response to Cd stress. The results provide theoretical guidance for remediation of Cd pollution using goldenrain tree.

## 2. Materials and Methods

### 2.1. Preparation of Materials

Seedlings of goldenrain tree were collected from Suqian, Jiangsu Province, China. Annual branches were cut to 25 cm lengths, in addition to branches with 2–3 lateral branches, green leaf blades, and roots of uniform length. Half of the Hoagland medium was hydroponic culture, the culture temperature was controlled at 24 ± 5 °C, the photoperiod was 12 h light/day, and the experiment was conducted after 45 days of culture. The stress condition was hydroponically cultured for 3 days in improved Hoagland nutrient solution with Cd supplied as CdCl_2_ at 0 mg/L (control), 25 mg/L, or 50 mg/L; the growth status is shown in Figure 1. The selection of Cd concentration refers to previous research results obtained by this research team [17] and takes the maximum tolerance concentration and the median value of *Koelreuteria paniculata*. After Cd stress, leaves and roots were collected, placed in 1.5 mL centrifuge tubes, frozen immediately in liquid nitrogen, and stored at −80 °C until use. In each treatment, there were three independent biological replicates of leaves and roots. Thus, 18 total samples included three replicates of the following six treatments (Y = leaves, gen = roots): YCd0, YCd25, YCd50, genCd0, genCd25, and genCd50.

### 2.2. Physiological Index of Cadmium Stress

Superoxide dismutase (SOD) activity was detected using the *p*-nitro blue tetrazolium chloride photochemical reduction method [18]. Peroxidase (POD) activity was determined using the guaiacol method [19]. Catalase (CAT) activity was determined using ultraviolet absorption spectrometry [20]. Malondialdehyde (MDA) content was determined using the glucosinolate barbituric acid colorimetric method [21].

### 2.3. RNA-Seq and Annotation

RNA isolation and purification and cDNA library construction and sequencing were conducted as described previously [22]. In brief, total RNA was extracted from leaves and roots using the CTAB method [23], and then quantity and quality were determined using a NanoDrop ND1000 spectrophotometer (NanoDrop Technologies, Wilmington, DE, USA) and an Agilent Bioanalyzer 2100 system (Agilent Technologies, Palo Alto, CA, USA), respectively. The integrity of RNA was examined on 1% agarose gels. Oligo (dT) magnetic beads were used to purify mRNA. A cDNA Synthesis Kit (TaKaRa, Tokyo, Japan) was used to synthesize cDNA, which was then linked with sequencing adapters [24]. Libraries were sequenced on an Illumina HiSeq 4000 platform (Illumina, San Diego, CA, USA). Unigene sequences obtained from the author’s laboratory transcriptome database by RSEM algorithm were integrated for annotation [22]. The whole set of annotated genes can be accessed in the National Center for Biotechnology Information SRA database (accession number PRJNA666052).

### 2.4. Differentially Expressed Genes

Reads were aligned and mapped to the reading of each gene by HTSeq (v0.5.4p3) [25] and then normalized to FPKM (fragments per kilobase of transcript per million mapped reads) following the method of Mao et al. [26]. Differentially expressed genes (DEGs) were recruited according to log_2_ (fold change) ≥ 1 and corrected *p* ≤ 0.005. Gene ontology (GO) enrichment was analyzed using GOseq (1.10.0) [27].

### 2.5. Reverse-Transcription Quantitative PCR Validation

Six transcription factor (TF) genes were selected for reverse-transcription quantitative PCR (RT-qPCR) confirmation. The RT-qPCR was performed using an ABI 7500 Fast Real-Time Detection System (Applied Biosystems, Carlsbad, CA, USA). The 2^−∆∆CT^ values from three technical replicates of each sample were used for statistical analysis. The significance level was set at *p* < 0.05.

## 3. Results

### 3.1. Growth, Antioxidant Activity, and MDA Content in Goldenrain Tree under Cadmium Stress

Cadmium stress significantly affected goldenrain tree growth (Figure 1). Compared with the control, leaf and root development decreased significantly at 50 mg/L CdCl_2_. Cadmium did not significantly affect SOD activity in leaves and roots (Figure 2A). However, compared with YCd0, leaf POD activity increased significantly (*p* < 0.05) in YCd50. Similarly, compared with genCd0, root POD activity increased significantly (*p* < 0.01) in genCd25 and genCd50 (Figure 2B). In both leaves and roots, CAT activity increased significantly in both Cd25 and Cd50 treatments (Figure 2C), compared with controls. Compared with controls, MDA content increased significantly in YCd50 (*p* < 0.001) and in genCd25 (*p* < 0.01) and genCd50 (*p* < 0.001) (Figure 2D).

### 3.2. Transcriptomic Analysis of Goldenrain Tree Leaves and Roots under Cadmium Stress

Global transcriptomic changes were examined in leaves and roots of goldenrain tree at 0, 25, and 50 mg/L Cd. A total of 65,171 genes were obtained with a GC percentage of 44.98% and Q30 quality scores ≥ 92.87%. The average length of assembled reads was 1121 bp, and the N50 length was 1631 bp (Table 1).

Differential expression of genes was analyzed with DESeq (*p* < 0.01) using FPKM to calculate gene expression levels. As shown in Figure 3, in comparisons with the control, more genes were downregulated than upregulated in roots in both 25 and 50 mg/L Cd treatments. By contrast, in leaves, more genes were upregulated than downregulated in both 25 and 50 mg/L Cd treatments, in comparison with the control.

### 3.3. GO Enrichment Analysis of Differentially Expressed Genes in Leaves and Roots

GO enrichment analysis was performed to categorize potential functions of DEGs, and the top 20 enriched GO terms of all DEGs are shown in Figure 4. In roots in the comparison between the control and 25 mg/L Cd, the main GO terms enriched in the cellular component category were intracellular ribonucleoprotein complex, ribonucleoprotein complex, and macromolecular complex. In the molecular function category, structural molecule activity, nucleoside triphosphatase activity and gene expression, and hydrogen ion transmembrane transport were enriched GO terms. In roots treated with 50 mg/L Cd (Figure 4B), the significantly enriched GO terms in the cellular component category were intrinsic component of membrane (GO:0031224), membrane (GO:0016020), and membrane part (GO:0044425).

In leaves, under low Cd stress at 25 mg/L, most DEGs activated in the cellular component category were enriched in intrinsic component of membrane (GO:0031224) and membrane part (GO:0044425) (Figure 4C). In the molecular function category, DEGs were primarily enriched in transferase activity, transferring phosphorus-containing (GO:0016772), and kinase activity (GO:0016301). In leaves under high Cd stress, DEGs in the in cellular component category were enriched in intrinsic component of membrane (GO:0031224) and membrane (GO:0016020) (Figure 4D). In the biological process category, the most enriched GO terms were response to organonitrogen compound (GO:0010243) and regulation of cell death (GO:0010941).

There were three replicates for each treatment. In the correlation matrix, each row corresponds to a module, and each column corresponds to one replicate of a treatment. The color of a cell at a row–column intersection indicates the correlation coefficient between the module and the sample. A high degree of correlation between a specific module and a sample is indicated by dark red (positive) or dark green (negative).

### 3.4. Identification of Co-Expression Modules

To investigate the gene regulatory network response to Cd stress, co-expressed gene sets were analyzed via weighted gene co-expression network analysis (WGCNA) [28]. Genes with high correlation coefficients, indicating high degrees of interconnection, were defined as modules. In Figure 5, each differently colored branch in the hierarchical cluster indicates a unique module, and each leaf in the tree represents one gene. The gene expression profile of each module was represented by its eigengene, the most notable component. As shown in Figure 5, there were 425 genes in the green module, with specific accumulation in YCd0. There were 634 genes in the brown module, accumulating specifically in YCd50, which indicated that this group of genes might be responsible for stress at the high Cd concentration. There were 65 genes in the midnight-blue module, with specific accumulation in YCd50, which indicated that this group of genes might be responsible for a response to low-concentration stress (Figure 5).

### 3.5. Transcription Regulatory Modules Associated with Antioxidant Enzymes and Malondialdehyde

Another WGCNA was performed to identify genes associated with SOD, POD, and CAT activities and MDA content (Figure 6). Analysis of the module–trait relations revealed that the blue module was highly correlated with activity of POD (*r* = 0.82, *p* = 3 × 10^−5^) (Figure 6A). The function of genes in the blue module was further examined, and Figure 6B shows the top 20 KEGG enrichment pathways. In the blue module, ribosome (939 genes) and phagosome (93 genes) pathways were significantly enriched. In the magenta module, 11 genes were highly correlated with MDA, and, in the KEGG enrichment analysis (Figure 6C), biosynthesis of secondary metabolites, glycolysis/gluconeogenesis, diterpenoid biosynthesis, alpha-linolenic acid metabolism, metabolic pathways, and phenylpropanoid biosynthesis pathways were significantly enriched. Many genes in the red module were highly correlated with CAT, and, in the KEGG analysis (Figure 6D), metabolic pathways, nitrogen metabolism, biosynthesis of amino acids, and 2-oxocarboxylic acid metabolism were significantly enriched.

### 3.6. Transcription Factor Responses to Cadmium Stress

Transcription factors are important in plant defense and stress response. Ten transcription factor families were differentially expressed in leaves and roots of goldenrain tree under Cd stress, including C2H2, MYB, WRKY, and bZIP (Figure 7A).

Among differentially expressed bZIP genes, nine of ten were highly expressed in genCd50 (Figure 7B). The bZIP-like gene Unigene0052331 was highly expressed in genCd25, genCd50, and YCd50. Of 31 differentially expressed C2H2 genes, 26 were highly enriched in genCd50 (Figure 7C). The C2H2-like Unigene0046162, Unigene0011683, and Unigene0046012 were highly expressed in YCd50. Unlike C2H2 and bZIP genes, expression of seven MYB genes gradually decreased with increase in Cd content (Figure 7D). In leaves, the MYB-like Unigene0019400 and Unigene0027656 were significantly expressed without Cd stress and expression decreased gradually with increased Cd content. In leaves, 11 WRKY genes (Unigene0040124, Unigene0058490, Unigene004348, Unigene0026078, Unigene0015499, Unigene0043630, Unigene006379, Unigene0042807, Unigene0045320, Unigene009358, and Unigene0032493) increased expression with the increase in Cd content (Figure 7E). In roots, only two genes (Unigene005624 and Unigene006755) increased expression with the increase in Cd stress (Figure 7E).

### 3.7. Validation of RNA-Seq by RT-qPCR

To validate RNA-seq data, six transcription factor genes were selected for RT-qPCR validation (Figure 8). Generally, gene expression levels and trends of regulation (increase/decrease) detected with RNA-seq and based on RT-qPCR were consistent with one another.

## 4. Discussion

### 4.1. Antioxidant Enzyme Activities and Malondialdehyde Content Increased Dramatically under Cadmium Stress

Cadmium is a common toxic heavy metal that threatens plant development and growth [29]. For example, lateral roots and total root volume decrease in soybean under Cd stress, compared with controls, and Zou et al. [30] found that leaf color lightens, biological yield decreases, and plants ultimately die under Cd stress, compared with controls. In the current study, Cd stress also negatively affected goldenrain tree development, and leaf and root growth decreased substantially at 50 mg/L CdCl_2_ (Figure 1).

Plants produce free radicals under stress that can destroy cell membranes through lipid peroxidation to generate MDA. Antioxidant enzymes, including SOD, POD, and CAT, are important in removing free radicals and active oxygen and, thus, preventing damage to cells [19]. Thus, both antioxidant enzymes and MDA are useful indicators of physiological responses to adverse conditions. In this study, POD and CAT activity and MDA content increased significantly at 50 mg/L, whereas SOD activity was not significantly affected by Cd treatment (Figure 2). Thus, cellular damage caused by Cd stress might explain the rapid and strong physiological changes in the leaves and roots of goldenrain tree.

### 4.2. Differentially Expressed Genes Responded Rapidly to Cadmium Stress

Plant responses to biotic and abiotic stresses are typically rapid. For example, transcriptome analysis revealed 2670 DEGs in soybean in response to Cd stress. Puente et al. [31] found that Cd exposure strongly increases transposon expression in green algae and that heavy metal stress induces oil biosynthesis genes in *Chlamydomonas*. In an analysis of global genome expression in plants exposed to 50 µM of Cd or lead, 65 genes were upregulated and 338 were downregulated with exposure to Cd, whereas 19 genes were upregulated and 76 were downregulated with exposure to lead [32]. Consistent with previous studies, many DEGs were identified under Cd stress in this study (Figure 3), with more than 600 genes involved in various physiological processes significantly up- or downregulated. Furthermore, in roots, intracellular ribonucleoprotein complex, ribonucleoprotein complex, and macromolecular complex were the primary enriched GO terms (Figure 4). Plants evolve strategies to survive and perform under many biotic and abiotic stresses that restrict plant productivity. However, maintaining protein functional conformation and preventing aggregation of non-native proteins, which leads to metabolic disruption, are of prime importance [33]. For example, heat-shock proteins as chaperones are essential in conferring plant tolerance to biotic and abiotic stresses. Thus, many of the DEGs might have been enriched in GO terms associated with intracellular ribonucleoprotein complex because, to overcome Cd stress, plants must produce many functional proteins.

Environmental stresses damage plant membranes. In response to heat stress, leaves specifically increase levels of galactolipids containing linoleate (18:2) in chloroplasts and phospholipids containing palmitate (16:0), stearate (18:0), and oleate (18:1) in the endoplasmic reticulum and plasma membrane [34]. Expression patterns and localizations of PIPs can also change in response to abiotic stresses [35]. In leaves of goldenrain tree, most DEGs activated by Cd stress were enriched in intrinsic component of membrane (GO:0031224) and membrane part (GO:0044425) (Figure 4).

### 4.3. Transcription Factors Were Important in the Response to Cadmium Stress

Transcription factors are important regulators in responses to stresses [36]. In particular, plant TFs C2H2, MYB, WRKY, and bZIP have key regulatory roles under heavy metal stress. For example, knockout of *thmea1,* a C2H2-like gene, in *Trichoderma* spp. results in 12.9% higher tolerance to copper than that in the wild type [37]. The C2H2-type zinc finger TF GhSTOP1 is essential for aluminum and proton stress tolerance and lateral root initiation in cotton [38]. In this study, 31 C2H2 genes were differentially expressed in goldenrain tree under Cd stress, of which 26 were highly enriched in genCd50. Among the C2H2-like genes, Unigene0046162, Unigene0011683, and Unigene0046012 were highly expressed in YCd50. These results indicated that *C2H2* genes were important in the response to Cd stress.

The MYB proteins are a large family of transcription factors that are important in plant stress responses. For example, *StMYB030* is significantly upregulated in soybean under salt and drought stress [39], and overexpression of *RsMYB1* increases heavy metal stress tolerance in transgenic *Petunia* [40]. In this study, seven MYB genes were highly expressed in the control (YCd0), but expression gradually decreased with increasing Cd concentration. Among these genes, Unigene0019400 and Unigene0027656 were significantly expressed, and their expression decreased gradually with the increase in Cd content. To explain the molecular mechanisms, further exploration is needed. In *Arabidopsis*, *MYB49* positively regulates the expression of *bHLH38* and *bHLH101*, leading to the activation of IRT1, a metal transporter involved in Cd uptake [41].

In *Zea mays*, Cd upregulated the expression of *ZmWRKY4* and the activities of SOD and ascorbate peroxidase [42]. In *Arabidopsis*, *WRKY13* activated the expression of *PDR8* to positively resist Cd tolerance [43]. The results in this study are consistent with those of previous studies, and the expression of 11 WRKY genes in leaves increased with increased Cd content. In roots, when Cd content increased, expression of Unigene005624 and Unigene006755 increased.

*Arabidopsis* with overexpressed *BnbZIP2* is sensitive to drought and Cd stress during seed germination [44]. Tian et al. [45] characterized the regulatory network of one member of the bZIP family, NCU03905, and found that it encodes an Ap1-like protein that is involved in resistance to multiple stress responses, including oxidative and heavy-metal stresses. In this study, nine of 10 differentially expressed bZIP genes were highly expressed in genCd50. Specific regulation mechanisms need to be explored further.

## 5. Conclusions

Cd stress significantly increased POD, and CAT activities and MDA content in leaves and roots of goldenrain tree. In transcriptome data, more than 2000 DEGs were identified that were significantly associated with the primary events of Cd exposure, including members of the C2H2, MYB, WRKY, and bZIP TF families. These results provide new information to direct further research on the molecular mechanisms of the TFs responding to Cd stress in goldenrain tree leaves and roots. This study provides reference for the resistance mechanism of woody plants, and woody plants have the prospect of being used as environmental remediation plants. The breeding of environmental remediation plants will definitely be further studied by means of molecular biology in the future. Goldenrain tree shows promise as an environmental remediation plant.

## Figures and Tables

**Figure 1 ijerph-18-12046-f001:**
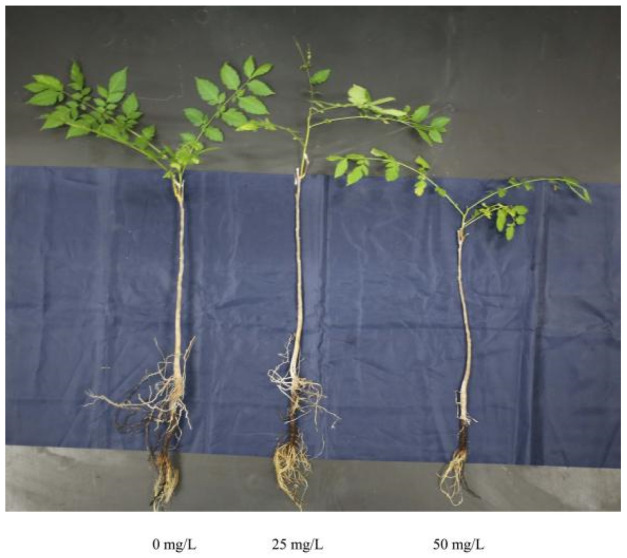
Phenotypes of the goldenrain tree *Koelreuteria paniculata* after cadmium treatment.

**Figure 2 ijerph-18-12046-f002:**
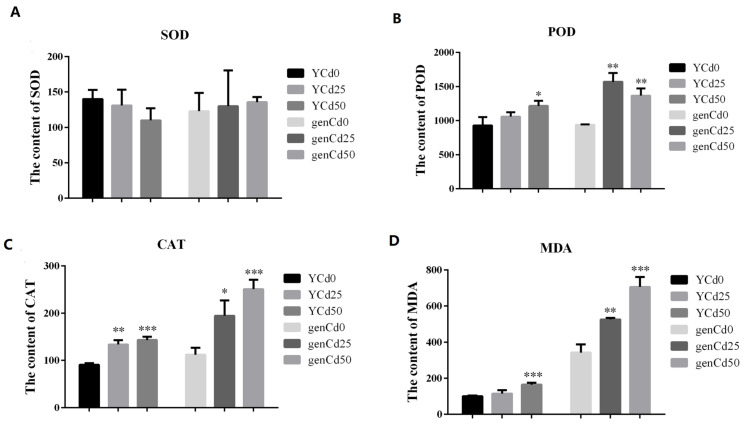
Activity of (**A**) SOD, (**B**) POD, and (**C**) CAT and content of (**D**) MDA in goldenrain tree treated with 0 mg/L, 25 mg/L, and 50 mg/L CdCl_2_ for 3 days. Y = leaves; gen = roots. Values are the mean ± SE (*n* = 3); * *p* < 0.05, ** *p* < 0.01, *** *p* < 0.001 (Tukey’s test).

**Figure 3 ijerph-18-12046-f003:**
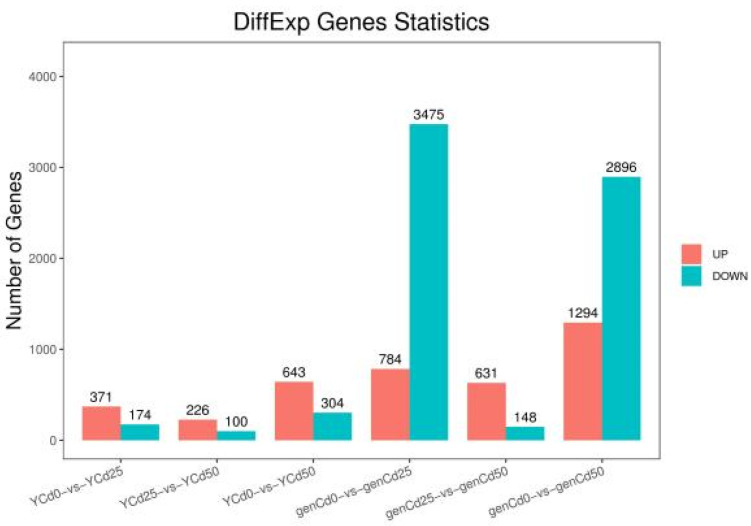
Histogram of differentially expressed genes that were either upregulated (Up) or downregulated (Down) in goldenrain tree treated with 0 mg/L, 25 mg/L, and 50 mg/L CdCl_2_ for 3 days. Y = leaves; gen = roots.

**Figure 4 ijerph-18-12046-f004:**
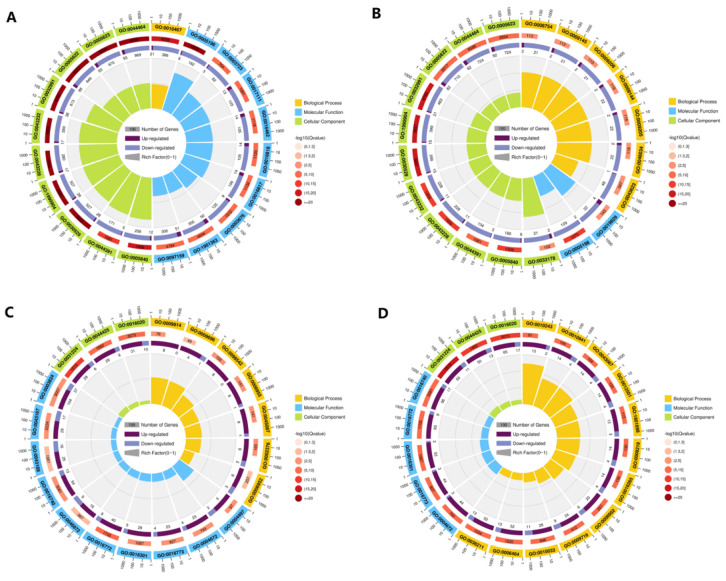
Gene Ontology (GO) enrichment analysis of differentially expressed genes in goldenrain tree treated with 0 mg/L, 25 mg/L, and 50 mg/L CdCl_2_ for 3 days. Y = leaves; gen = roots. (**A**) genCd0 vs. genCd25; (**B**) genCd0 vs. genCd50; (**C**) YCd0 vs. YCd25; (**D**) YCd0 vs. YCd50. The first outer circle contains the top 20 enriched GO terms, and different colors represent different ontologies (biological process, molecular function, and cellular component). The second circle includes the number and Q-value of the GO term in the background gene. The more genes there are, the longer the bar; the smaller the Q-value is, the deeper the red color. The third circle shows bar charts of upregulated (dark purple) and downregulated (light purple) genes, with the specific values at the bottom. The wedges in the fourth circle show the rich factor value of each GO term.

**Figure 5 ijerph-18-12046-f005:**
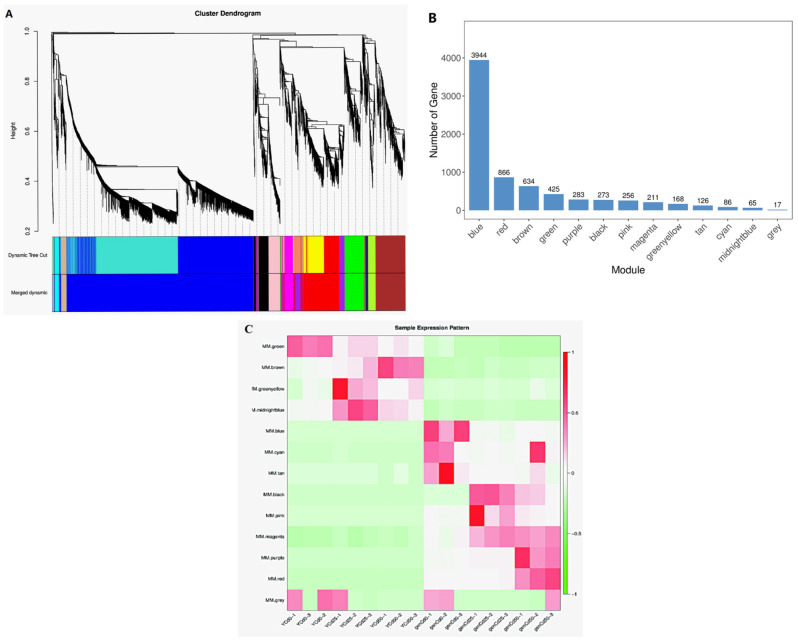
Co-expression network in goldenrain tree under cadmium stress. (**A**) Hierarchical cluster tree showing co-expression modules identified by weighted gene co-expression network analysis. Each leaf in the tree represents one gene. (**B**) Gene number in each module from (**A**). (**C**) Module–sample association for goldenrain tree treated with 0 mg/L, 25 mg/L, and 50 mg/L CdCl2 for 3 days. Y = leaves; gen = roots.

**Figure 6 ijerph-18-12046-f006:**
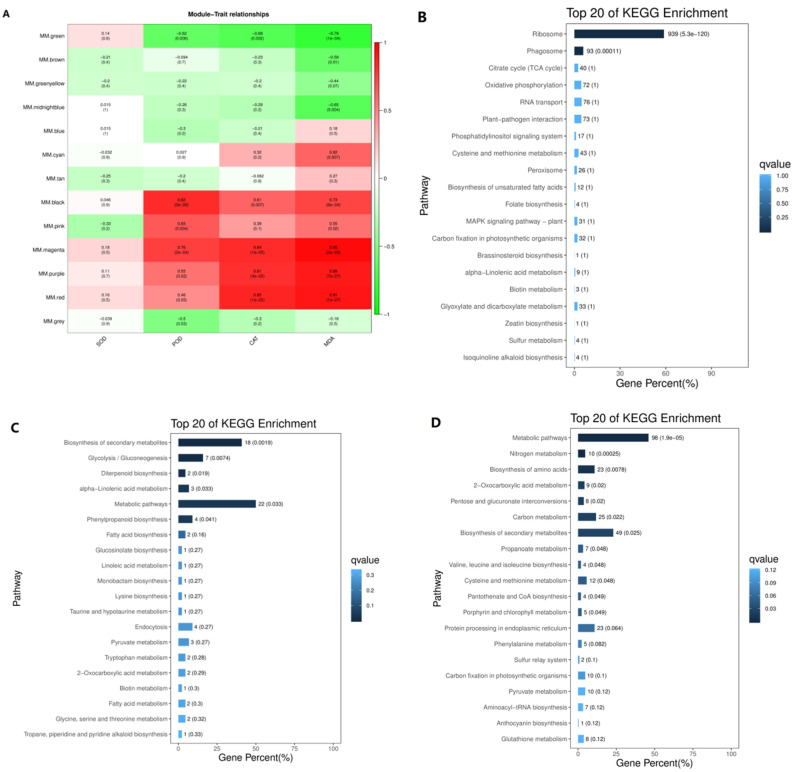
Co-expression network and KEGG enrichment pathways in goldenrain tree under cadmium stress. (**A**) Module-trait association. In the correlation matrix, each row corresponds to a module, and each column corresponds to one trait. The color of a cell at a row–column intersection indicates the correlation coefficient between the module and the trait. A high degree of correlation between a specific module and a trait is indicated by dark red (positive) or dark green (negative). (**B**) Top 20 KEGG enrichment pathways in the blue module. (**C**) Top 20 KEGG enrichment pathways in the magenta module. (**D**) Top 20 KEGG enrichment pathways in the red module.

**Figure 7 ijerph-18-12046-f007:**
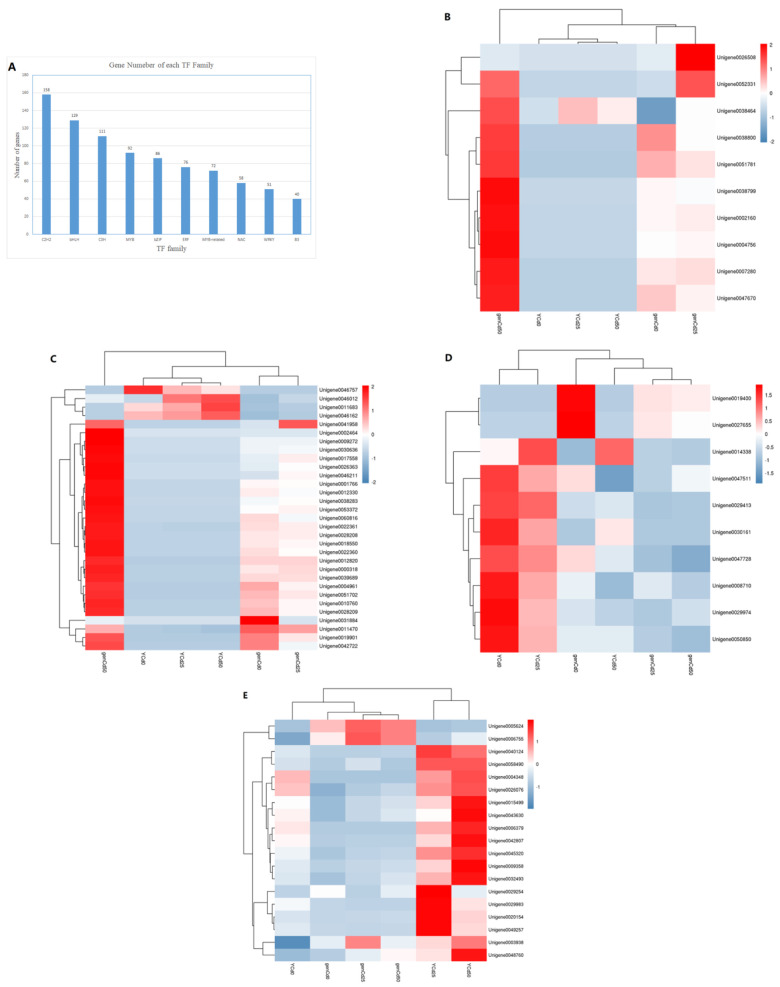
Transcription factor (TF) family analysis of goldenrain tree treated with 0 mg/L, 25 mg/L, and 50 mg/L CdCl_2_ for 3 days. Y = leaves; gen = roots. (**A**) Gene number in different TF families. (**B**–**E**) Heat maps of expression profiles of (**B**) bZIP, (**C**) C2H2, (**D**) MYB, and (**E**) WRKY genes.

**Figure 8 ijerph-18-12046-f008:**
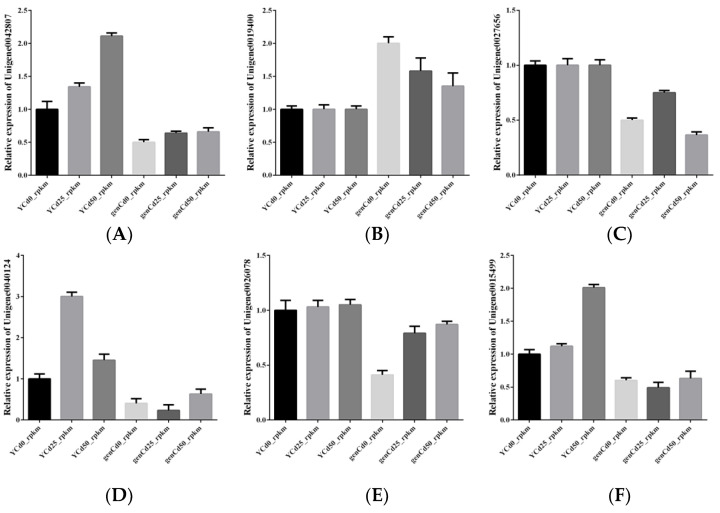
RT-qPCR results of six transcription factors: (**A**) Unigene0042807; (**B**) Unigene0019400; (**C**) Unigene0027656; (**D**) Unigene0040124; (**E**) Unigene0026078; (**F**) Unigene0015499.

**Table 1 ijerph-18-12046-t001:** Assembly quality statistics of RNA-seq of goldenrain tree.

Gene Number	GC Percentage	N50 Number	N50 Length	Max Length	Min Length	Average Length	Total Assembled Bases
65,171	44.9824	13,438	1631	16,907	201	1121	73,081,674

## Data Availability

The data that support the findings of this study are available from the corresponding author upon reasonable request.

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
