# Peer review of "Transcriptome Profiles of Leaves and Roots of Goldenrain Tree (Koelreuteria paniculata Laxm.) in Response to Cadmium Stress"

_ijerph, 2021, doi:10.3390/ijerph182212046_

Round 1

Reviewer 1 Report

  • Please include the economic importance of the golden rain trees in the introduction.
  • Include clear-cut objectives of the study in the Introduction section in light of research gaps.
  • Screening protocol references are required.
  • The conclusion can sum up research findings in light of research gaps and strategies for future research

Author Response

Comments from Reviewer #1:

  • Please include the economic importance of the golden rain trees in the introduction.

Author response:Thank you for your valuable suggestion. According to your suggestion, we have inserted the following paragraph in Line74-76 of Page 2:

“Golden rain tree has traditionally been used for landscaping because of its beautiful yellow flowers. In recent years, some people have tried to use its seeds to produce biodiesel[14], which has high economic value.

  • Include clear-cut objectives of the study in the Introduction section in light of research gaps.

Author response:Thank you for your valuable suggestion. According to your suggestion, we have inserted the following paragraph in Line82-83 of Page 2:

“To explore its molecular response mechanism under Cd stress, reveal its metabolic pathway and differential genes in response to Cd stress”

  • Screening protocol references are required.

Author response:Thank you for your comments.The selection of Cd concentration refers to the previous research results of this research team[15]and takes the maximum tolerance concentration and the median value of Koelreuteria paniculata. I have added relevant references.

Reference:

[15] Yang, L.P., Zhu, J., Wang, P., Zeng, J., Tan, R., Yang, Y. Z., Liu Z.M. Effect of Cd on growth, physiological response, cd subcellular distribution and chemical forms of koelreuteria paniculata.[J]. Ecotoxicology Environmental Safety, 2018, 160:10-18. doi:10.1016/j.ecoenv.2018.05.026

  • The conclusion can sum up research findings in light of research gaps and strategies for future research

Author response:Thank you for your valuable suggestion. According to your suggestion, we have inserted the following paragraph in Line353-364of Page 12:

To summarize, Cd stress significantly increasedPOD, and CAT activities and MDA content in leaves and roots of golden rain tree. In transcriptome data, more than 2,000 DEGs were identified that were significantly associated with the primary events of Cd exposure, including those of members of C2H2, MYB, WRKY, and bZIP TF families. These results provide new information to direct further research on the molecular mechanisms of the TFs responding to Cd stress in golden rain tree leaves and roots.This study provides reference for the resistance mechanism of woody plants, and woody plants have the prospect of being used as environmental remediation plants. The breeding of environmental remediation plants will definitely be further studied by means of molecular biology in the future. Golden rain tree has the prospect of environmental remediation plants.

Reviewer 2 Report

Please check comments in the main text.

Author Response

Comments from Reviewer #2:

I think it would better to change title

as you said increase biochemical but not recognize which biochemical parameters you mean, so it is unclear to say in general only biochemical activity.

One more it is obvious any stress make increase or decrease biochemical activities in plant but some of them are positive and helpful for plant to increase tolerance against stress condition, but some of them have negative effects so the title is unclear in my oponion.

Author response:Thank you very much for your suggestions,and we change the title to Transcriptome profiles of goldenrain tree(Koelreuteria paniculata Laxm)leaves and roots in response to cadmium stress

Please improve introduction by some new references which published recently such as: https://doi.org/10.1016/j.plaphy.2021.08.013

https://doi.org/10.1016/j.ecoenv.2020.111887

https://doi.org/10.1080/15226514.2020.1801573

Author response:Thank you for your valuable suggestion. According to your suggestion, we have inserted the following paragraph in Line55-61 of Page 2:

Cadmium accumulation in plants may cause physiological, biochemical and structural changes. Accumulation of cadmium can change nutrient absorption of minerals, inhibit stomatal opening, disrupt Calvin cycle enzymes, photosynthesis, carbohydrate metabolism, change antioxidant metabolism and reduce crop productivity through interaction with plant water balance [7]. A large number of studies have shown that the main action site of Cd is the biosynthesis of photosynthetic pigments, especially chlorophyll and carotenoids, and plant growth[8-9]”

Please add more detail about plant growth conditions

Author response:Thank you for your suggestion

Seedlings of golden rain tree were collected from Suqian, Jiangsu Province, China. Annual branches were cut to 25-cm lengths, branches with 2-3 lateral branches, leaf blade green, roots of uniform length.Half of hoagland medium was hydroponic culture, the culture temperature was controlled at 24 ±5℃, the light time was 12h/d, and the experiment was conducted after 45 days of culture. Stress condition is  hydroponically cultured for 3 d in improved Hoagland nutrient solution with Cd supplied as CdCl2 at 0 mg/L (control), 25 mg/L, or 50 mg/L,the growth status is shown in Fig 1. The selection of Cd concentration refers to the previous research results of this research team[15]and takes the maximum tolerance concentration and the median value of Koelreuteria paniculata. After Cd stress, leaves and roots were collected, placed in 1.5-mL centrifuge tubes, frozen immediately in liquid nitrogen, and stored at −80°C until use. In each treatment, there were three independent biological replicates of leaves and of roots. Thus, 18 total samples included three replicates of the following six treatments (Y = leaves, gen = roots): YCd0, YCd25, YCd50, genCd0, genCd25, genCd50.” has been inserted in line 87-100

why only these enzyme?

Did you have any reason to select only these parameters?

Author response:Thank you for your comments. These enzymes are oxidative reactive kinases involved in the heavy metal poisoning reaction of plants. The determination of SOD activity and POD activity and CAT activity can indirectly reflect the damage degree of membrane system and the stress resistance of plants. Studies have shown that the changes of their enzyme activity can reflect both the sensitivity of plants to heavy metals and the strength of resistance.

why started with figure??????

Author response:Thank you for your comments.We have adjusted the position of the figure.

Why only MDA?

It think it would be much better if you have data for H2O too.

Author response:Thank you for your comments.At present, there are three common methods for the determination of H2O2 in plants. The disadvantage of confocal laser detection using fluorescence probe is that the specificity is not high, and other intracellular reactive oxygen species can also be measured. The shortcomings of DAB histochemical staining and UV spectrophotometer are low sensitivity. We have determined the content of H2O2, the use of the existing conditions of our laboratory ultraviolet spectrophotometer detection, H202 data is not very accurate no sense, so there's no reflected in the article, we think that the determination of MDA typical relative H2O2 is stable, can to a certain extent, reflect the environmental stresses of plant damage.

missing the significant letter????

Author response:Thank you for your suggestion.We have modified it

please replace with better and clear caption

Author response:Thank you for your suggestion.We changed to a better caption.

Reviewer 3 Report

Editorial corrections.

  1. in lines 21 and 22: the digit “2” in the “CdCl2” compound should be entered as subscript
  2. species names in lines 19, 339, 345, 354, 363, 364, 369, 390, 393, 422 and 427 should be written in italics
  3. in lines 363, 364, 372, and 377, Authors should remove the multispaces
  4. in line 255, "uM" unit should be replaced by "µM"
  5. in line 86 one abbreviation of "RNA" should be deleted

Substantive content of publication:

  1. The introduction should contain more information on the pollution of the environment with cadmium compounds. There is information that this pollution is a serious problem, but for me, as a researcher from outside of China, this information is insufficient. In my opinion, the introduction to the subject of the reviewed publication would be more complete if the Authors provided some data showing the real level of contamination of the water-soil environment with this metal. This would additionally allow to assess whether the applied concentrations of cadmium chloride (and, in fact, of cadmium itself) correlate with the level of cadmium in the environment.
  2. Could the authors add to the introduction data on the maximum concentrations of cadmium (or other heavy metals) to which the golden rain tree is resistant (if such data are available in the literature)?
  3. In the statement opening the “Conclusions” chapter: "(...) Cd stress significantly increased SOD, (...)" (line 311) there is some inconsistency to what was written earlier regarding to superoxidase dismutase activity. In the chapter “3.1. Growth, antioxidant activity, and MDA content in golden rain tree under cadmium stress” (line 113 in “Results” chapter) and in the chapter “4.1. Antioxidant enzyme activities and malondialdehyde content increased dramatically under cadmium stress” (line 247 in the "Discussion" chapter), Authors clearly indicated that SOD activity was not significantly affected by Cd treatment. Thus, the sentence presented in the "Conclusions" chapter must be modified so that the conclusions were consistent with the previously discussed results.

Author Response

Comments from Reviewer #3:

Editorial corrections.

  1. in lines 21 and 22: the digit “2” in the “CdCl2” compound should be entered as subscript
  2. species names in lines 19, 339, 345, 354, 363, 364, 369, 390, 393, 422 and 427 should be written in italics
  3. in lines 363, 364, 372, and 377, Authors should remove the multispaces
  4. in line 255, "uM" unit should be replaced by "µM"
  5. in line 86 one abbreviation of "RNA" should be deleted

Author response:Thank you for pointing out my mistakes, We have completed the modification

Substantive content of publication:

  1. The introduction should contain more information on the pollution of the environment with cadmium compounds. There is information that this pollution is a serious problem, but for me, as a researcher from outside of China, this information is insufficient. In my opinion, the introduction to the subject of the reviewed publication would be more complete if the Authors provided some data showing the real level of contamination of the water-soil environment with this metal. This would additionally allow to assess whether the applied concentrations of cadmium chloride (and, in fact, of cadmium itself) correlate with the level of cadmium in the environment.

Author response:Thank you for your valuable suggestion. According to your suggestion, we have inserted the following paragraph in Line50-54 of Page 2:

A previous report showed that as much as 88 tonnes of cadmium was released into the environment in 1989 by the non-ferrous metallurgy industry[5]. In industrial regions, the mean concentrations of Cd are 79.2 times greater than the Grade II environmental quality standard for soils in China (GB15618-1995)(About 20 mg/Kg)[6].

  1. Could the authors add to the introduction data on the maximum concentrations of cadmium (or other heavy metals) to which the golden rain tree is resistant (if such data are available in the literature)?

Author response: It has been added to the method “The selection of Cd concentration refers to the previous research results of this research team[15]and takes the maximum tolerance concentration and the median value of Golden Rain Tree. ”

  1. In the statement opening the “Conclusions” chapter: "(...) Cd stress significantly increased SOD, (...)" (line 311) there is some inconsistency to what was written earlier regarding to superoxidase dismutase activity. In the chapter “3.1. Growth, antioxidant activity, and MDA content in golden rain tree under cadmium stress” (line 113 in “Results” chapter) and in the chapter “4.1. Antioxidant enzyme activities and malondialdehyde content increased dramatically under cadmium stress” (line 247 in the "Discussion" chapter), Authors clearly indicated that SOD activity was not significantly affected by Cd treatment. Thus, the sentence presented in the "Conclusions" chapter must be modified so that the conclusions were consistent with the previously discussed results.

 Author response: Thank you very much for pointing out my mistake. Although the SOD activity of Koelreuteria paniculata increased to some extent under cadmium stress, it did not significantly increase.I have changed line367-368 to:

To summarize, Cd stress significantly increased POD, and CAT activities and MDA content in leaves and roots of golden rain tree.